# Clinical Tick-Borne Encephalitis in a Roe Deer (*Capreolus capreolus* L.)

**DOI:** 10.3390/v14020300

**Published:** 2022-01-31

**Authors:** Graziana Da Rold, Federica Obber, Isabella Monne, Adelaide Milani, Silvia Ravagnan, Federica Toniolo, Sofia Sgubin, Gianpiero Zamperin, Greta Foiani, Marta Vascellari, Petra Drzewniokova, Martina Castellan, Paola De Benedictis, Carlo Vittorio Citterio

**Affiliations:** 1U.O. Ecopathology SCT2-Belluno, Istituto Zoprofilattico Sperimentale delle Venezie (IZSVe), Via Cappellari 44/A, 32100 Belluno, Italy; fobber@izsvenezie.it (F.O.); ccitterio@izsvenezie.it (C.V.C.); 2OIE Collaborating Centre for Diseases at the Animal/Human Interface, Istituto Zooprofilattico Sperimentale delle Venezie (IZSVe), Viale dell’Università 10, 35020 Legnaro, Italy; imonne@izsvenezie.it (I.M.); amilani@izsvenezie.it (A.M.); sravagnan@izsvenezie.it (S.R.); ftoniolo@izsvenezie.it (F.T.); ssgubin@izsvenezie.it (S.S.); gzamperin@izsvenezie.it (G.Z.); gfoiani@izsvenezie.it (G.F.); mvascellari@izsvenezie.it (M.V.); pdrzewniokova@izsvenezie.it (P.D.); mcastellan@izsvenezie.it (M.C.); pdebenedictis@izsvenezie.it (P.D.B.); 3Laboratory for Viral Genomics and Trascriptomics, Istituto Zooprofilattico Sperimentale delle Venezie (IZSVe), Viale dell’Università 10, 35020 Legnaro, Italy; 4Laboratory of Parasitology Micology and Sanitary Enthomology, Istituto Zooprofilattico Sperimentale delle Venezie, Viale dell’Università 10, 35020 Legnaro, Italy; 5Histopathology Laboratory, Istituto Zooprofilattico Sperimentale delle Venezie, Viale dell’Universita 10, 35020 Legnaro, Italy; 6Laboratory for Viral Emerging Zoonoses, Istituto Zooprofilattico Sperimentale delle Venezie, Viale dell’Università 10, 35020 Legnaro, Italy

**Keywords:** roe deer, tick-borne encephalitis, neurologic disease, pathology, genetic characterization

## Abstract

*Tick-borne encephalitis virus* (TBEV) is the causative agent of tick-borne encephalitis (TBE), a severe zoonosis occurring in the Palearctic region mainly transmitted through *Ixodes* ticks. In Italy, TBEV is restricted to the north-eastern part of the country. This report describes for the first time a case of clinical TBE in a roe deer (*Capreolus capreolus* L.). The case occurred in the Belluno province, Veneto region, an area endemic for TBEV. The affected roe deer showed ataxia, staggering movements, muscle tremors, wide-base stance of the front limbs, repetitive movements of the head, persistent teeth grinding, hypersalivation and prolonged recumbency. An autopsy revealed no significant lesions to explain the neurological signs. TBEV RNA was detected in the brain by real-time RT-PCR, and the nearly complete viral genome (10,897 nucleotides) was sequenced. Phylogenetic analysis of the gene encoding the envelope protein revealed a close relationship to TBEV of the European subtype, and 100% similarity with a partial sequence (520 nucleotides) of a TBEV found in ticks in the bordering Trento province. The histological examination of the midbrain revealed lymphohistiocytic encephalitis, satellitosis and microgliosis, consistent with a viral etiology. Other viral etiologies were ruled out by metagenomic analysis of the brain. This report underlines, for the first time, the occurrence of clinical encephalitic manifestations due to TBEV in a roe deer, suggesting that this pathogen should be included in the frame of differential diagnoses in roe deer with neurologic disease.

## 1. Introduction

*Tick-borne encephalitis virus* (TBEV), the causative agent of the severe, and even lethal, zoonosis tick-borne encephalitis (TBE), is a member of the genus *Flavivirus*, family Flaviviridae; the viral genome is a positive-sense, single-stranded RNA molecule of about 11 Kb. Nowadays, five subtypes of TBEV are known, phylogenetically classified and characterized by different geographical distribution and severity of disease: European (TBEV-Eu), Siberian (TBEV-Sib), Far Eastern (TBEV-FE), Baikalian (TBEV-BKl) and Himalayan (TBEV-Him) [1,2]. The prevailing subtype in Western Europe is TBEV-Eu, the least virulent subtype with a case fatality rate of less than 2% compared to the TBEV-FE (20–60%) and TBEV-Sib (7–8%) [3,4].

TBEV is mainly transmitted through bites of infected ticks of the genus *Ixodes* to different mammals, including humans [5,6]. Occasionally, human infection can also occur via alimentary route in people consuming unpasteurized milk from infected dairy cattle or small ruminants [7]. *Ixodes ricinus* is the main vector of TBEV-Eu, while TBEV-Sib and TBEV-FE are mainly transmitted by *Ixodes persulcatus*. Ticks act as the vector and a reservoir of TBEV, which is transmitted from viraemic competent host animals to ticks [8] or by infected to uninfected ticks co-feeding on the same host [9,10,11]. Notably, infected ticks can transmit the virus to their offspring through transovarian transmission [12].

The ecology and epidemiology of TBEV depend on different interconnected factors, such as climate, landscape and density of ticks and their hosts, including both species that are competent for TBEV transmission (as small rodents) and other species (e.g., wild ungulates) playing a key role in maintaining a consistent tick population, although not competent for viral transmission [13]. For this reason, natural foci of TBEV have a patchy distribution, ranging from a few square metres to several square kilometres [14,15].

The actual impact of TBEV in mammals other than humans is poorly investigated. Indeed, few reports of clinical TBE in animals are available in the literature [16,17,18,19,20,21,22,23,24] and the infection is notably asymptomatic in ruminants, with few exceptions [25,26,27,28]. No clinical cases have been reported in roe deer (*Capreolus capreolus* L.) so far, although this species is one of the most abundant cervids in Europe and acts as a preferential host for *I. ricinus* [29,30]. In roe deer, viraemia is generally low and short, so that it is considered unable to infect ticks [31,32,33,34,35,36]. On the other hand, serological monitoring in this species has been used as a sentinel for TBE foci detection [36].

In the present paper, we describe clinical signs, pathological findings and the viral characterization in a case of TBE in a roe deer in Belluno province (Veneto region, North- Eastern Italy), a longtime recognized endemic area, accounting for about 40% of all the Italian cases of TBE in humans.

## 2. Materials and Methods

### 2.1. Case History, Clinical Signs and Autopsy Findings

On 2 June 2021, the Belluno Provincial Police found a one-year-old female roe deer in the Belluno municipality, in a location named Modolo (46°07′58.0′′ N; 12°15′11.3′′ E; 390 m a.s.l.). The animal was in poor general condition and showed various neurological signs including ataxia, staggering movements, muscle tremors, wide-base stance of the front limbs, repetitive movements of the head, persistent teeth grinding, hypersalivation and prolonged recumbency (videos are provided as Appendix A). Due to the severity of the clinical manifestation and the passive surveillance plans undertaken in the region to rule out infectious diseases of public health interest, the animal was, therefore, humanely culled and promptly submitted to the Istituto Zooprofilattico Sperimentale delle Venezie (IZSVe) for post-mortem examination and diagnosis.

An autopsy revealed no significant lesions to explain the neurological signs. Besides evident teeth wear, presumably due to grinding, other findings were nonspecific. Concerning the brain, only mild vascular congestion was observed.

Seventeen engorged ticks, morphologically identified as adult *I. ricinus*, were found in the skin and collected for molecular analyses to assess for the presence of common pathogens of public health interest. Roe deer’s brain was collected for virological and histological investigations.

### 2.2. Virologic and Molecular Testing

First, rabies virus infection was ruled out through the direct fluorescent antibody (DFA) test coupled with real time PCR (RT-PCR) and virus isolation attempt [37,38]. A iQ Check *Listeria monocytogenes* II kit (Certificate number: BRD 07/10–04/05) was also used to detect DNA of *L. monocytogenes*. Based on negative results, the presence of a Flavivirus infection was, therefore, investigated through a specific RT-PCR targeting the 3′ noncoding region of the TBEV genome [39].

Nucleic acids were extracted from tick samples using the All Prep DNA/RNA mini Kit (Qiagen, Valencia, CA, USA), according to the manufacturer’s instructions. DNA was amplified by SYBR Green real-time PCR (rPCR) assays for *Borrelia burgdorferi* (s.l.), *Rickettsia* spp., *Babesia* spp., and *Anaplasma phagocytophilum*; RNA was amplified by a specific real-time PCR (rRT-PCR) for TBEV [39,40].

### 2.3. Illumina MiSeq Sequencing, Bioinformatics Analysis and Phylogeny

Total RNA was extracted from brain tissue with the QIAamp Viral RNA mini kit (Qiagen, Valencia, CA, USA) and then subject to double stranded cDNA synthesis using the Maxima H Minus Double-stranded cDNA synthesis kit #K2561 (ThermoFisher, Waltham, MA, USA), purified with magnetic beads (Agencourt AMPure XP, Beckman Coulter, Brea, CA, USA), and quantified with the Qubit dsDNA HS assay kit (ThermoFisher, Waltham, MA, USA). Library preparation was performed using a Nextera XT DNA sample preparation kit and processed on an Illumina MiSeq instrument with a MiSeq reagent kit V3 (2 × 300 bp paired-end [PE] mode; Illumina, San Diego, CA, USA)

Illumina reads quality was assessed using FastQC v0.11.2 (https://www.bioinformatics.babraham.ac.uk/projects/fastqc) and consensus sequence of the whole genome obtained using the pipeline, as detailed in Appendix B.

Whole genome consensus sequence and the nucleotide portion coding for the envelope protein (E) were compared with the most related sequences available in GenBank database. A multiple nucleotide sequences alignment for E containing the aforementioned sequences (21RS1767 and blast results), as well as representative ones for the five subtypes described so far TBEV-Eu, TBEV-Sib, TBEV-FE, TBEV-Bkl and TBEV-Him was obtained using MAFFT v. 7 [2,41,42,43,44]. A maximum likelihood phylogenetic tree was obtained using GTR + F + I + G4. To assess the robustness of individual nodes of the phylogeny, 1000 bootstrap replicates were performed. A phylogenetic tree was visualized using FigTree v1.4.3.

Envelope protein (E) and nonstructural protein 5 (NS5) amino acid sequences have been inspected to look for major viral determinants of virulence previously identified [45].

### 2.4. Histological Examination, Immunohistochemistry and Immunofluorescence

A sample of midbrain collected at autopsy was fixed in 10% neutral buffered formalin, processed routinely, embedded in paraffin, microtome-sectioned, stained with hematoxylin and eosin (H&E) and mounted on glass slides for histologic examination under optic microscope. Immunohistochemistry (IHC) for *Listeria monocytogenes* and *Toxoplasma gondii* was performed on further 4 µm formalin-fixed paraffin-embedded sections in an automated immunostainer (Discovery Ultra; Roche, Ventana Medical Systems™, Oro Valley, AZ, USA). After dewaxing, sections were submitted to antigen retrieval with ULTRA Cell Conditioning Solution (pH 6.0, Ventana) at 91 °C (24–32 min). Sections were incubated with polyclonal rabbit antibodies, anti-Listeria O (1:500 diluted for 32 min, BD Difco™, Franklin Lakes, NJ, USA) and anti-*Toxoplasma gondii* inflammatory profilin (1:50 diluted for 40 min, Clinisciences™, Nanterre, France) at room temperature. The OmniMap anti-rabbit HRP (Ventana, Santa Clara, CA, USA) was used as detection system. Sections of bovine and feline brain naturally infected by *Listeria monocytogenes* and *Toxoplasma gondii*, respectively, were used as positive controls. Replicate tissue sections, submitted to the same protocol without the primary antibody, were used as negative controls.

For immunofluorescence analysis of the active microglia in the roe deer midbrain, 4 µm-thick sections were re-hydrated and antigen retrieval was performed by incubation in citrate buffer 0.01 M pH 6 at 95 °C for 20 min. Slides were then permeabilized for 20 min at RT with PBS 1% Triton X-100. Slides were saturated with blocking buffer (BSA 5% in PBS 0.1% Triton [PBSt]) for 1 h and incubated overnight at 4 °C with the mouse monoclonal anti-Iba1 (ionized-calcium binding protein 1) primary antibody (1:100 diluted, Abcam, Cambridge, UK). The following day, samples were incubated in the dark 2 h with a secondary antibody (goat anti-mouse Alexa Fluor 568, Thermo Fisher Scientific, Waltham, MA USA) conjugated with a fluorophore, previously diluted in 1% BSA in PBSt. Slides were washed and mounted in with Fluoroshield Mounting Medium with DAPI (Sigma-Aldrich, Saint Louis, MO, USA), to label cell nuclei. Images were acquired with Leica TCS SP8 confocal microscope equipped with a CCD camera using LAS AF 2.7.3.9723 software and analyzed using ImageJ.

## 3. Results

Molecular investigations performed on the brain yielded positive results for TBEV. The metagenomic approach allowed us to reveal and identify TBEV as the unique viral pathogen in the brain. An almost complete genome sequence with a length of 10,897 nucleotides was obtained (sequences available in GenBank under the accession numbers OM084948). The coding region extended from the nucleotide position 114 to 10,358 and corresponded to a polyprotein of 3414 amino acids. From the BLAST carried out with the consensus sequence of the genome towards the sequences available in the database on 22 November 2021 (https://blast.ncbi.nlm.nih.gov/Blast.cgi), the highest similarity (99.17%) was identified with the TBEV AS33 (GQ266392) strain isolated from ticks in Germany in 2005. As previously seen for the German strains named Salem (FJ572210) and AS33 (GQ266392), the virus genome presents a deletion of 43 nucleotides in the untranslated region (NTR) of polyadenosine in relation to reference strain Neudoerfl (U27495) [45]. At the amino acid level, there were 18 differences scattered throughout the genome compared to the AS33 strain, while there were 37 compared to the Neudoerfl strain. Amino acids differences observed from the comparison of E and NS5 proteins of the 21RS1767, AS33 and Neudoerfl viruses are summarized in Appendix A (Neudoerfl numbering for nucleotide positions was considered). Interestingly, E protein showed three differences in antigenic domains I and II [46]; in particular, 21RS1767 showed Glu (E) at position 331 of the central domain (I) and Thr (T) at position 361 of the dimerization domain (II), similar to the Neudoerfl virus, whereas, at position 408 (central domain (I)), it showed Ile (I), similar to the AS33 virus. At position 761 in the E stem anchor, the 21RS1767 virus showed an Ile (I), whereas AS33 and Neudoerfl showed a Leu (L). NS5 protein showed a total of ten differences; in particular, five were within the N-terminal RNA methyltransferase (MTase) domain and one in the RdRp catalytic domain. Position 2532 and 2559 showed Arg (R) and Glu (E), respectively, similar to Neudoerfl, whereas 2562, 2619 and 2764 showed the same residues present in AS33. Position 3297 in the RdRp catalytic domain showed a Val (V) for both 21RS1767 and AS33, whereas Neudoerfl showed Ala (A). Mutations in the E and NS protein involved in TBE neuroinvasiveness and neurovirulence previously listed in the review of the Kellman et al. (2018) have not been detected [3].

Based on phylogenetic analysis of gene E, sample 21RS1767 grouped within the European subtype (Figure 1); the highest percentage of nucleotide similarity (100%) was found to be with the partial sequence (520 nt) of a TBEV obtained from a pool of ticks collected in 2018 on Monte Calisio in Val d’Adige (TN) (Genbank accession number MN746771), about 100 km from Modolo, which also presented the Ile (I) at position 408 [46].

TBEV and *Babesia* spp. were not found in ticks. Four ticks out of seventeen harbored *A. phagocytophylum,* two tested positive for *R. helvetica* and two were co-infected by two pathogens: one by *A. phagocytophilum* and *R. helvetica* and another by *A. phagocytophilum* and *B. afzelii*.

At histological examination, the midbrain showed a moderate, multifocal encephalitis characterized by perivascular cuffs and neuropil infiltrates of lymphocytes and histiocytes mixed with fewer eosinophils and rare neutrophils, in both grey (Figure 2) and white matter. Moreover, gliosis, neuronal chromatolysis and rare microglial nodules were observed in the grey matter. The presence of active microglia was further confirmed through immunofluorescence staining for the marker Iba1 (Figure 3). The increased number of active immune cells in the midbrain of the infected roe deer has been observed by comparing the number of Iba1 cells in a non-infected, control roe deer brain.

Diagnostic tests for *L. monocytogenes* and *T. gondii* yielded negative results.

## 4. Discussion

We herewith described the clinical, pathological and virological findings of a symptomatic roe deer infected by a TBEV from a known endemic area of North-Eastern Italy. The TBEV responsible for this case was characterized as a European subtype with the highest percentage of nucleotide similarity (100%) at the *E* gene with the partial sequence (520 nt) of a TBEV (MN746771) obtained from a pool of ticks collected in Trento province in 2018, a location about 100 km from the case described herein [46].

Histological findings in the roe deer’s midbrain were supportive of a neurotropic virus infection, although we did not observe any extensive neuronal necrosis and neuronophagy as described in human or canine TBE [47]. Notably, we observed the activation of microglia in the midbrain of the infected animal, as depicted by histological analysis and immunofluorescence for the specific marker Iba1. Unfortunately, we did not find any viral protein by immunofluorescence, as the specimen material was too small and probably not representative of the ongoing TBEV infection that was actually identified by real-time RT-PCR. Given the lower presence of eosinophils and neutrophils in the infiltrate, we further ruled out the presence of co-infections through immunohistochemistry tests for *T. gondii* and *L. monocytogenes*.

Molecular investigations did not reveal the presence of mutations in the E and NS protein notably involved in TBE neuroinvasiveness and neurovirulence previously identified [3,48]. It is, however, worth mentioning the presence of amino acid differences compared to AS33 and Neudoerfl strains in the antigenic domains (I and II), a region involved in the viral membrane fusion with the host cell, a finding deserving further investigation.

Our report leads to the inclusion of TBEV in the frame of a differential diagnosis of clinical encephalitis in roe deer, especially within and around known foci. In a public health perspective, the case herein described, as well as those quoted in the literature, cannot be used as an alert. Actually, even in the case of the emergence of TBEV, clinical episodes in animals, if present, would be, unfortunately, preceded by far by cases in humans [49]. Nevertheless, cases in animals should be monitored and framed in a consistently mutating ecopathological scenario. In fact, like other large wild and domestic mammals, roe deer do not seem to play a direct role in the maintenance of TBEV, because, generally, the level of viraemia after natural infection is too low to infect ticks and, therefore, the absence of TBEV in the ticks collected from the roe deer’s carcass was not surprising [50,51]. Nevertheless, roe deer play a role in the TBEV ecology, being a key host in granting survival and abundance of *I. ricinus* populations [52]. It is worth noting that, in North-Eastern Italy, and, more broadly, in the Alps, ungulates have been increasing for decades, in parallel to the progressive depopulation and loss of human activities. Since fawns and yearlings (as the case presented herein) are the age categories that, more than others, account for an incomplete immunity to microparasites such as TBEV and heavy infestation by all active stages of *I. ricinus* [13], and since studies in other species (e.g., horses) suggest declining levels of TBEV antibodies following passive transfer from foals to yearlings, a high roe deer population turnover in conjunction with environmental conditions favorable to ticks may lead to unpredictable variations in the ecology of the disease [53]. Considering all the above, although referring to a single case, our report stresses the importance to flank information on host population dynamics to pathogen knowledge.

## Figures and Tables

**Figure 1 viruses-14-00300-f001:**
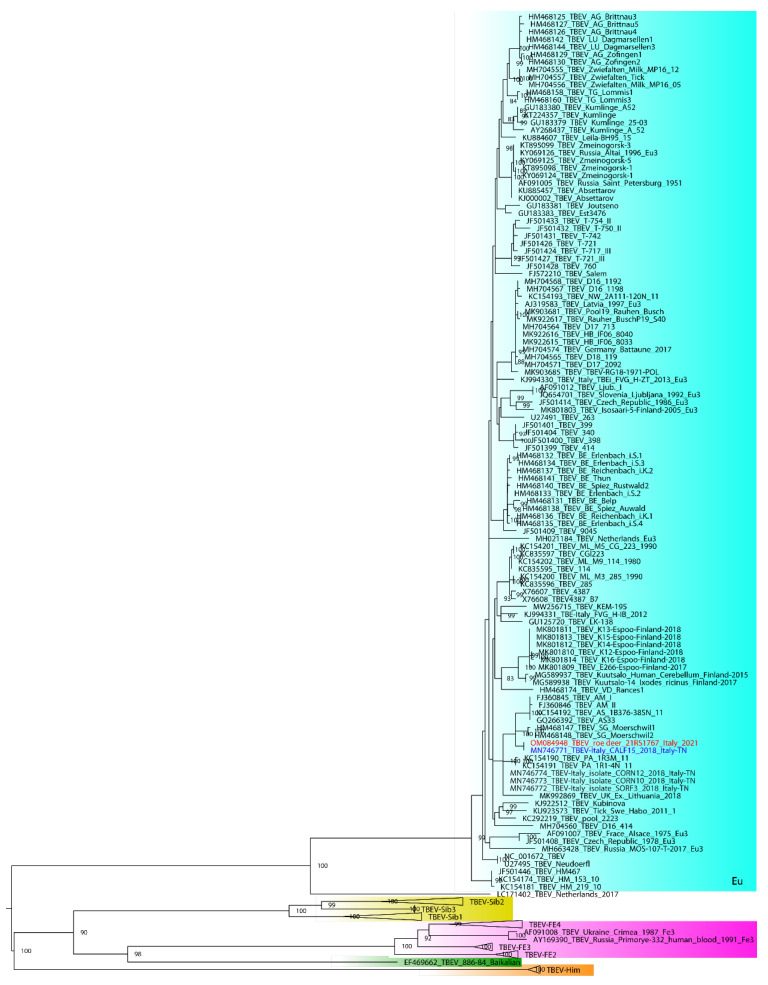
Maximum likelihood phylogenetic tree of TBEV *E* gene. In red the virus OM084948_TBEV_roe deer_21RS1767_Italy_2021 amplified from the roe deer’s brain. European (TBEV-Eu), Siberian (TBEV-Sib), Far Eastern (TBEV-FE), Baikalian (TBEV-Bkl) e Himalayan (TBEV-Him) are grouped and highlighted, respectively, in ligth blue, yellow, purple, green and orange. The highest percentage of nucleotide similarity (100%) was found with the partial sequence (520 nt) of a TBEV obtained from a pool of ticks collected in 2018 on Monte Calisio in Val d’Adige (TN) (virus marked in blue).

**Figure 2 viruses-14-00300-f002:**
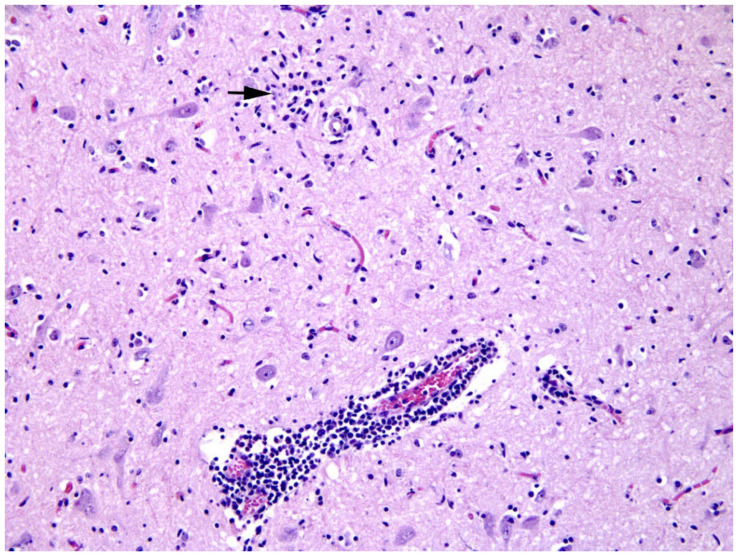
Histologic section of the midbrain (grey matter) stained with H&E. A perivascular space (lower center) is infiltrated by lymphocytes and histiocytes; the surrounding neuroparenchyma is hypercellular due to increased numbers of glial cells, including microglia, that occasionally form nodules (arrow) or arrange near the body of neurons (satellitosis).

**Figure 3 viruses-14-00300-f003:**
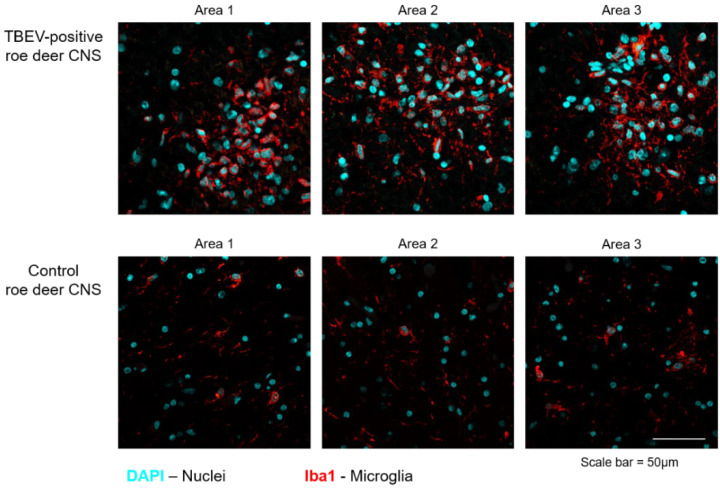
Representative staining of Iba1+ cells (microglia, red) in TBEV-positive and control roe deer midbrain. Nuclei are stained with DAPI, light blue. Scale bar: 50 μm.

## Data Availability

Consensus sequence of TBEv was deposided in GenBank under accession number OM084948. MiSeq raw sequencing data are available at the NCBI Sequence Read Archive (SRA) under accession number PRJNA793165.

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
