# Peer review of "Clinical Tick-Borne Encephalitis in a Roe Deer (Capreolus capreolus L.)"

_viruses, 2022, doi:10.3390/v14020300_

Round 1

Reviewer 1 Report

Dear authors, please find the review suggestions below.

Discussions included in the results section please pass them to the discussion chapter (which is too weak).

There are too many references. Select only those that are relevant to your research.

Author Response

Dear Reviewer,

We would like to thank you for your useful suggestions that helped us to improve the quality of our manuscript. We have carefully taken into consideration all of your comments.

As suggested , we strenghtended the Discussion chapter including additional comments. We are aware that references may appear too much for a short communication of single case. At the same time, however, we think that the novelty of this report could give the opportunity to the Reader to have a brief review of TBEV ecology and epidemiology.

Sincerely yours.

Graziana Da Rold

Reviewer 2 Report

Please see comments in the attached document.

Author Response

Dear Reviewer,

We would like to thank you very much for your appreciation and even more for your careful and detailed review. All your suggestions have been accepted, allowing to definitely improve the manuscript.

As for the request of detecting a TBEV antigen, histological findings in the roe deer’s CNS were supportive of a neurotropic virus infection, and we also observed the activation of microglia in the midbrain of the infected animal, as depicted by histological analysis and immunofluorescence for the specific marker Iba1. Unfortunately, we were not able to detect any TBEV protein by immunofluorescence; we would like to point out the difficulty in finding viral proteins in the CNS of the animal under consideration, as the specimen material we had was too small (a piece of the animal midbrain) and probably not representative of the ongoing TBEV infection that was actually identified by real-time RT-PCR.

Sincerely yours.

Graziana Da Rold

Reviewer 3 Report

The authors describe a case of symptomatic TBE in a roe deer. The authors state that this is the first case of clinical overt TBE infection in a roe deer observed and described so far. The manuscript describes clearly the methodology and presents the results. Although I have no doubt on the etiology of the clinical symptoms I suggest to add some additional data to improve the whole story:

a picture with positive immunohistochemistry (as described in chapter 2.4; line 209) but not shown. Instead a picture with Iba1+ cells is shown. However there is no discussion about these results.

In Figure 2 the staining is missing (although I think it was HE staining, this should be added).

In Figure 3 the authors state that this is from TBEV positive roe deer CNS. It would be nice if they could show this using direct/indirect immunofluorescence and staining some positive cells and show in the same part.

It would be nice if the authors could discuss shortly why ticks were negative although the roe deer had an TBE infection.

If possible it would be increase the value of the publication if the authors could add a serological result on TBE antibodies of the roe deer which would finally show the immunological response to the infection and be another good prove.

The authors intensively discuss the phylogenetic relation with the ticks in Monte Calisii/Val d'Adige. Maybe a second phylogenetic tree (just using the partial sequence mentioned for the E genes of some representative EU subtype strains) is added showing this relationship as in the tree presented these data are not visible.

Author Response

Dear Reviewer,

We would like to thank you for your useful suggestions that helped us to improve the quality of our manuscript. We have carefully taken into consideration all of your comments.

As for the request of detecting a TBEV antigen, histological findings in the roe deer’s CNS were supportive of a neurotropic virus infection, and we also observed the activation of microglia in the midbrain of the infected animal, as depicted by histological analysis and immunofluorescence for the specific marker Iba1. Unfortunately, we were not able to detect any TBEV protein by immunofluorescence; we would like to point out the difficulty in finding viral proteins in the CNS of the animal under consideration, as the specimen material we had was too small (a piece of the animal midbrain) and probably not representative of the ongoing TBEV infection that was actually identified by real-time RT-PCR.

Concerning why ticks were negative although the roe deer had an TBE infection, we reported in the discussion that it is attributable to the low viraemia after natural infection in wild ungulates, that is deemed unable to infect ticks. We could hypothesize that in a deer with clinical encephalitis, the viraemic phase has been over, but this would be too speculative. Moreover, in our experience, it has been extremely difficult to detect TBEV in ticks from animal hosts as well as from the environment (by flagging/dragging collection), even in logtime known foci in humans. Unfortunately, this makes surveys on ticks almost useless as a routinary surveillance tool for this zoonosis.

Concerning serological results on TBE antibodies, we fully agree with you that this would have been a valuable information. In our humble opinion, however, dealing with a single fatal case, with no opportunity to repeat analyses and in the absence of reference data, the results could be difficult to frame in a clinical perspective. For sure, instead, serology has a great importance in an epidemiological monitoring perspective. As an example, in 2006-2010 we performed a serological survey in regularly hunted roe deer across the southern Belluno province: the area around Modolo consistently showed high seropositivity for TBE in all the considering hunting seasons, ranging from 44% to 70% prevalence.

Finally, information about TBEV from Monte Calisio (TN) has been highlighted in the phylogenetic tree.

Sincerely yours.

Graziana Da Rold

Round 2

Reviewer 3 Report

There are still some minor type errors in the manuscript:

L94: signs

L96: concerning

L143: Glass

L284 stages

In Fig 1, the phylogenetic position of the Italian sequences MN746772, MN746773, MN746772 are unclear

Author Response

Dear Reviewer,

We would like to thank you for your useful suggestions that helped us to improve the quality of our manuscript.

As suggested , we corrected minor type errors at lines 94-96-143-284 and we submitted the article to mothertongue language editor.

In Fig. 1 we marked in blue the sequence from Italy with higher similarity to the roe deer sequence (OM084948). We decided to not highlight the other italian sequences (MN746772, MN746773, MN746772) because their nucleotide sequence similarity with OM084948 is lower.

Sincerely yours.

Graziana Da Rold